# Spt5 interacts genetically with Myc and is limiting for brain tumor growth in *Drosophila*

Julia Hofstetter[1],* , Ayoola Ogunleye[2],* , André Kutschke[1], Lisa Marie Buchholz[2], Elmar Wolf[1] , Thomas Raabe[3] , Peter Gallant[2]

The transcription factor SPT5 physically interacts with MYC oncoproteins and is essential for efficient transcriptional activation of MYC targets in cultured cells. Here, we use *Drosophila* to address the relevance of this interaction in a living organism. Spt5 displays moderate synergy with Myc in fast proliferating young imaginal disc cells. During later development, Spt5-knockdown has no detectable consequences on its own, but strongly enhances eye defects caused by Myc overexpression. Similarly, Spt5-knockdown in larval type 2 neuroblasts has only mild effects on brain development and survival of control flies, but dramatically shrinks the volumes of experimentally induced neuroblast tumors and significantly extends the lifespan of tumor-bearing animals. This beneficial effect is still observed when Spt5 is knocked down systemically and after tumor initiation, highlighting SPT5 as a potential drug target in human oncology.

## Introduction

Expression of MYC oncogenes is deregulated in most human tumors. Up to 28% of all tumors exhibit gene amplification of one of the *MYC* isoforms (*MYCN*, *MYCL* or *MYC*), defining *MYC* genes as the most frequently amplified oncogene family across human cancers (Schaub et al, 2018). Indeed, MYC is a crucial driver of tumorigenesis as demonstrated by mouse experiments involving MYC-overexpression (Adams et al, 1985; Kortlever et al, 2017), genetic depletion of endogenous (Sansom et al, 2007; Walz et al, 2014) or exogenous MYC (Jain et al, 2002), and expression of a dominant-negative variant of MYC (Soucek et al, 2008). MYC can therefore be considered a priority target for cancer therapy (Dang, 2012). At the same time, it is very challenging to target MYC directly, because it lacks enzymatic activity and probably pockets for small molecules (Nair & Burley, 2003). Instead, it seems possible to identify binding partners which the oncogenic function of MYC is fully dependent on, and to target

them, for example, the histone–methyl–transferase adapter protein WDR5 (Thomas et al, 2015; Lorenzin et al, 2016). In recent years, several additional MYC binding partners were identified by proteomic approaches, and MYC was shown to partake in multiple nuclear protein complexes (Koch et al, 2007; Buchel et al, 2017; Dingar et al, 2018; Kalkat et al, 2018; Baluapuri et al, 2019). To be considered as suitable for pharmaceutical targeting, such MYC binding partners should be (i) essential for MYC-driven oncogenic growth and (ii) dispensable for the integrity and proliferation of healthy tissue. The former is relatively easy to analyze systematically in transplantation-based murine tumor models (Vo et al, 2016), but the latter is very elaborate and expensive to study in mice. We therefore started to develop a *Drosophila* model to (i) validate the genetic interaction between MYC and its binding partners in vivo and (ii) to estimate effects on healthy tissue of animals and thus the potential therapeutic window.

The *Drosophila* genome encodes a single MYC homolog that accomplishes the functions of its vertebrate counterparts in normal cells, and it also acts as an oncogene in *Drosophila* tumor models. Here, we focused on a brain tumor model derived from neural stem cells (type II neuroblasts = NB II), which allows to study proliferation and tumorigenesis during brain development. Briefly, NB II produces intermediate neural progenitors (INPs) with a restricted proliferation potential, which in turn generate ganglion mother cells as the precursors of neurons and glia cells (Homem & Knoblich, 2012). NB II express the cell fate determinant brain tumor (Brat) and pass it on to their progeny (Bello et al, 2006; Betschinger et al, 2006; Lee et al, 2006). In case of brat mutations, INPs acquire NB II characteristics, resulting in large transplantable tumors (Bowman et al, 2008; Xiao et al, 2012; Janssens et al, 2014; Komori et al, 2018; Hakes & Brand, 2019). Brat belongs to the TRIM-NHL family of proteins which regulate gene expression by reducing translation and causing degradation of multiple mRNAs (Tocchini & Ciosk, 2015; Connacher & Goldstrohm, 2021). Brat targets many mRNAs involved in NB self-renewal, including Myc (Betschinger et al, 2006; Loedige et al, 2015). We exploited this tumor model to address the potential for interfering with tumor formation by targeting Myc interaction partners.

[1]Cancer Systems Biology Group, Theodor Boveri Institute, Biocenter, University of Würzburg, Würzburg, Germany   [2]Department of Biochemistry and Molecular Biology, Theodor Boveri Institute, Biocenter, University of Würzburg, Würzburg, Germany   [3]Molecular Genetics, Biocenter, Am Hubland, University of Würzburg, Würzburg, Germany

Correspondence: elmar.wolf@uni-wuerzburg.de; thomas.raabe@uni-wuerzburg.de; peter.gallant@uni-wuerzburg.de
*Julia Hofstetter and Ayoola Ogunleye are co-first authors

As a proof of the target validation concept, we chose the MYC binding partner SPT5. First, SPT5 was detected as a binding partner of both MYC (Baluapuri et al, 2019) and MYCN (Buchel et al, 2017), indicating that the interaction between MYC proteins and SPT5 is evolutionary conserved. Second, recombinantly expressed MYC and SPT5 build stable dimeric complexes in vitro, demonstrating their direct interaction (Baluapuri et al, 2019). Third, SPT5 is essential for MYC-mediated transcriptional activation, which is considered a key oncogenic function of MYC (Baluapuri et al, 2019). A function of SPT5 in transcription was already evident upon its initial discovery in a pioneer genetic screen by Winston and colleagues in yeast. Several suppressors of Ty (SPT) genes, including SPT5, were discovered, because their mutation reactivated the transcription of an auxotrophy gene that was silenced by proximal insertion of a Ty transposon (Winston et al, 1984). Subsequent work demonstrated direct interaction of SPT5 with SPT4 in yeast (Swanson et al, 1991; Hartzog et al, 1998) and the function of the mammalian SPT4/5 complex as a pausing factor named DSIF (DRB sensitivity-inducing factor) (Wada et al, 1998). SPT5 binds RNA polymerase II (RNAPII) and promotes transcriptional elongation and termination (Shetty et al, 2017; Henriques et al, 2018; Parua et al, 2018, 2020; Cortazar et al, 2019; Hu et al, 2021; Fong et al, 2022) and RNAPII processivity (Fitz et al, 2018) by binding to its DNA exit region, facilitating rewinding of upstream DNA and preventing aberrant back-tracking of RNAPII (Bernecky et al, 2017; Ehara et al, 2017). SPT5 homologues are found in all domains of life. SPT5 shares the N-terminal (NGN) and one KOW domain with its bacterial homolog NusG, but the eukaryotic protein contains several copies of the KOW domain and additional N- and C-terminal sequences (Yakhnin & Babitzke, 2014). Although SPT5 is an essential protein, its interaction with MYC could indicate that tumor cells are more dependent on the full function of SPT5 than untransformed cells.

Here, we explored the functional interaction between Myc and Spt5 in vivo in Drosophila and analyzed the consequences of Spt5 depletion in brain tumors induced by brat knockdown. We demonstrate a clear genetic interaction between Myc and Spt5 in developing eyes and a functional role of Spt5 in neuroblast proliferation. Strikingly, systemic knockdown of Spt5 from late larval stages onwards inhibits tumorigenesis, but is tolerated by normal tissue and massively extends the life span of tumor prone flies. This demonstrates not only that SPT5 is an attractive candidate for targeting MYC-mediated oncogenic growth, but also suggests that inhibition of an essential process, such as transcription, could open a therapeutic window in tumor treatment.

## Results

### Genetic interaction of Spt5 and Myc in Drosophila

The Drosophila genome encodes a single SPT5 homolog (Kaplan et al, 2000), which is 50% homologous to human SPT5 and contains all identified protein domains (Fig 1A). To investigate its genetic interaction with Myc we focused on adult eye phenotypes, which are known to be highly sensitive to alterations in Myc levels. Myc overexpression in post-mitotic cells of this tissue (using GMR-GAL4;

Fig 1B) induced excessive growth and apoptosis, resulting in oversized and aberrantly shaped adult eyes (Montero et al, 2008; Secombe et al, 2007; Steiger et al, 2008; Figs 1C and S1A and B). Spt5-levels were manipulated by expression of an siRNA targeting Spt5, or by overexpression of a mutated Spt5 cDNA that codes for WT Spt5 protein but is not recognized by the siRNA; ubiquitous expression of these transgenes in wandering larvae (using a heat-shock activated GAL4-driver; see the Materials and Methods section) altered Spt5 transcript levels to 62% ± 24% and 750% ± 90% of control, respectively. When driven with GMR-GAL4 in control eye discs, siRNA-mediated Spt5-knockdown had no discernible effect on adult eyes (Figs 1C and D and S1C). Knockdown of Spt5 in the Myc-overexpression context however dramatically altered eye morphology leading to a glassy surface, suggestive of apoptotic cell loss and ensuing fusion of neighboring ommatidia (Figs 1C and S1D). This phenotype was fully penetrant and accompanied by a reduction in overall eye size (Fig 1D). Importantly, this effect was not because of experimental off-target artefacts because it was completely rescued by expression of the siRNA-resistant Spt5 (Figs 1C and S1G and H). Overexpression of the siRNA-resistant Spt5 itself showed effects neither in control nor in Myc-overexpressing flies (Figs 1C and S1E and F). Together, these observations demonstrate that Myc and Spt5 functionally interact and suggest that the output of supraphysiological Myc levels is strongly influenced by Spt5 levels: whereas Myc-induced overgrowth is abrogated upon depletion of Spt5, Myc-induced apoptosis is potentiated. Similar observations were previously made for Myc's dimerization partner Max whose depletion in developing eyes also eliminated Myc-dependent overgrowth, but did not impair Myc-dependent apoptosis (Steiger et al, 2008).

Next, we addressed the organismal role of Spt5 during development. As described for yeast, Spt5 is an essential gene and Spt5 homozygous mutant flies do not reach adulthood (Mahoney et al, 2006). Spt5 heterozygotes were largely normal, except for a small but statistically significant reduction in adult body weight (Fig 2A). Such a weight defect was also described for hypomorphic $Myc^{P0}$ mutants, which additionally showed a slight delay in development (Johnston et al, 1999). The combination of both mutations did not affect the Myc-dependent developmental delay (Fig S2A), but resulted in a synergistic weight loss (BLISS score 14, SynergyFinder; Fig 2A). In addition, such doubly mutant flies had deformed eyes (not shown). Such an eye defect was not observed in either single mutant alone, but had previously been described as a typical manifestation of the strong genetic interaction between Myc and its partner RUVBL1/pontin (Bellosta et al, 2005).

The synergy between Spt5 and Myc in proliferating cells became even more evident when Spt5 and Myc levels were reduced specifically in developing eye imaginal discs (Fig 2B). In line with earlier publications, partial loss of Myc (to 20% of control level—see Wu & Johnston [2010]) in this system impaired growth of eye imaginal disc cells and resulted in smaller adult eyes made up of smaller ommatidia (Figs 2C and D and S2B–I; Bellosta et al, 2005). Combination of the partial loss of Myc with Spt5-knockdown showed clear synergy (BLISS score 16, SynergyFinder) and nearly eliminated eye development. These observations confirm a functional collaboration between Spt5 and Myc in the control of cellular growth and proliferation.

**A**

KOW1:

| | | |
|---|---|---|
| Human | 1 | LKPKSWVRLKRGIYKDDIAQVDYVEPS | 27 |

LK K WVRLKRG+YKDDIAQVDYV+ +

| Drosophila | 2 | LKVKQWVRLKRGLYKDDIAQVDYVDLA | 28 |

KOW2:

| Human | 1 | FQPGDNVEVCEGELINLQGKILSVDG | 26 |

F  GDNVEVC G+L NLQ KI+++DG

| Drosophila | 2 | FSMGDNVEVCVGDLENLQAKIVAIDG | 27 |

KOW3:

| Human | 1 | FKMGDHVKVIAGRFEGDTGLIVRVEENFVIL | 31 |

FK GDH +V+AGR+EG+TGLI+RVE    V+L

| Drosophila | 1 | FKTGDHARVLAGRYEGETGLIIRVEPTRVVL | 31 |

KOW4:

| Human | 1 | IHVKDIVKVIDGPHSGREGEIRHLFRSFAFLHCK | 34 |

I  +D+VKV++GPH+GR GEI+HL+RS AFLHC+

| Drosophila | 1 | IRRRDVVKVMEGPHAGRSGEIKHLYRSLAFLHCR | 34 |

KOW5:

| Human | 1 | ELIGQTVRISQGPYKGYIGVVKDATESTARVELH | 34 |

E++G+T++IS GPYKG +G+VKDATESTARVELH

| Drosophila | 1 | EILGKTIKISGGPYKGAVGIVKDATESTARVELH | 34 |

NGN:

| Human | 1 | DPNLWTVKCKIGEERATAISLMRKFIAYQFTDTPLQIKSVVAPEHVKGYIYVEAYKQTHV | 60 |

DPNLW VKC+IGEE+ATA+ LMRK++ Y  TD PLQIKS++APE VKGYIY+EAYKQTHV

| Drosophila | 1 | DPNLWMVKCRIGEEKATALLLMRKYLTYLNTDDPLQIKSIIAPEGVKGYIYLEAYKQTHV | 60 |

| Human | 61 | KQAIEGVGNLRLGYWNQQMVPIKEMTDVLKVVKE | 94 |

K  I+ VGNLR+G W Q+MVPIKEMTDVLKVVKE

| Drosophila | 61 | KTCIDNVGNLRMGKWKQEMVPIKEMTDVLKVVKE | 94 |

**B**

+/- Myc-OE    +/- Myc-OE

+/- Spt5-KD    +/- Spt5-KD
+/- Spt5-OE    +/- Spt5-OE

**C**

Ctr    Spt5-KD    Spt5-OE    Spt5-KD+OE

Ctr

Myc-OE

Ctr    Spt5-KD    Spt5-OE    Spt5-KD+OE

**D**

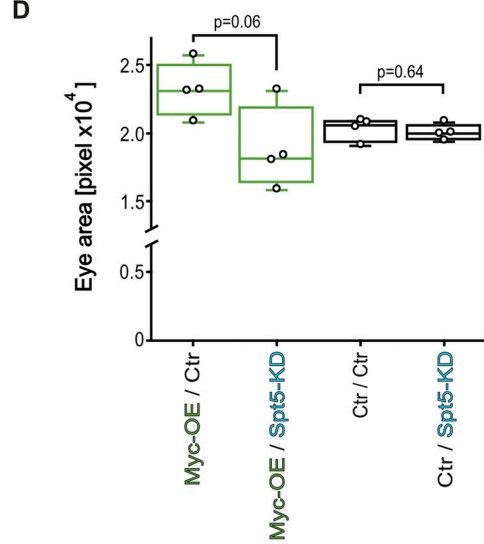

Figure 1. Genetic interaction of Spt5 with overexpressed Myc.
**(A)** Alignment of *Drosophila melanogaster* and human Spt5 proteins over all identified domains. **(B)** Scheme depicting GMR-GAL4–dependent transgene expression in differentiating eye imaginal disc cells from the second half of the third larval instar onward. GAL4 activates expression of a Myc cDNA and/or an Spt5 siRNA and/or an Spt5 cDNA (coding for WT Spt5 protein, but refractory to siSpt5). **(C)** Representative pictures of adult eyes of the indicated genotypes. **(D)** Quantification of the eye areas from control (black) or Myc-overexpressing (green) flies. Median adult eye size from four independent flies each. *P*-value was calculated using unpaired *t* test.

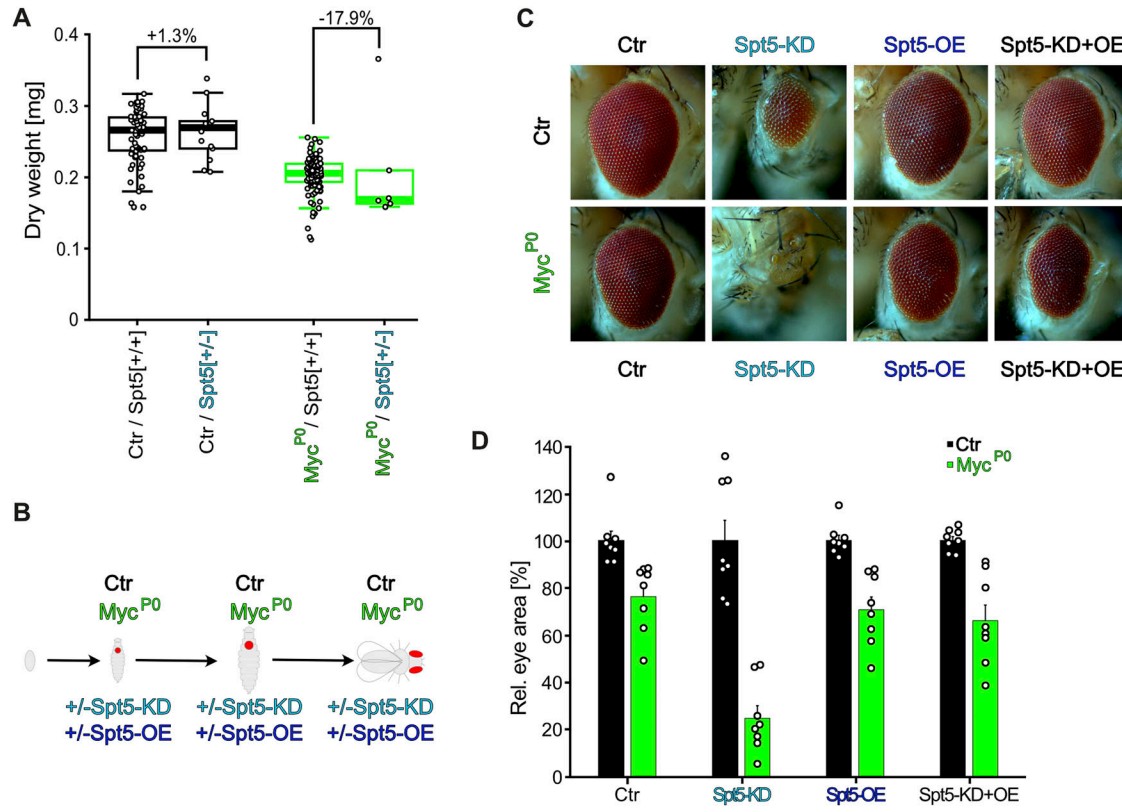

**Figure 2. Genetic interaction of Spt5 with a hypomorphic Myc mutant.**
**(A)** Median dry weight of adult Spt5[+/+] or Spt5[+/−] males (n = 6–109), in a Myc[wildtype] ("Ctr," black) or Myc[P0] (green) background. *P*-values were calculated using an unpaired *t* test. **(B)** Scheme illustrating the genetic manipulation: a ubiquitously expressed Myc cDNA was eliminated specifically in eye imaginal discs throughout larval development, thereby exposing the hypomorphic Myc[P0] allele or Myc[wildtype] ("Ctr"), whereas simultaneously driving Spt5 overexpression and/or knockdown (see the Materials and Methods section). **(C)** Representative pictures of adult eyes. **(D)** Quantification of eye areas, normalized in each case to the area of the matching Myc[wildtype] flies ("Ctr," black); n = 8 flies per genotype.

## Effect of Spt5 on NB II-tumor development

Having confirmed the importance of Spt5 for Myc-dependent physiological processes, we set out to explore the role of Spt5 in brain tumors that were induced by knockdown of the tumor suppressor brat specifically in larval NB II. The adult brains of Brat-knockdown animals are enlarged with a massive increase of cell number in the cortex region and a complete disruption of neuropil structures (Fig 3A and B). In contrast, knockdown of Spt5 in NB II had only minor effects on adult brain structures, for example, resulting in a ventral opening of the ellipsoid body of the central complex, which is one descendant of NB II cell lineages. Simultaneous knockdown of Spt5 and Brat abrogated the overgrowth phenotype and largely restored normal brain structures (Fig 3B). To quantify this effect, we expressed luciferase in the cells experiencing Brat knockdown. Luminometry of total lysates from young adults confirmed the strong growth-suppressive effect of Spt5-knockdown specifically in tumorous animals as opposed to control animals; expression of the siSpt5-insensitive *Spt5* transgene abrogated the effects of siSpt5, demonstrating its specificity (Fig 3C). Consistent with these findings, brat knockdown led to a massive expansion of NB II cell lineages, which was largely abolished by simultaneous Spt5-knockdown (Fig S3A).

To study the underlying cellular differences between the different genotypes, we analyzed NB II lineages in third instar larval brains by concurrent expression of GFP and stainings for Deadpan (Dpn) and Asense (Ase), which are transcription factors that serve as markers for neuroblasts: NB II (of which there are eight per brain hemisphere) express Dpn but not Ase (Dpn+ Ase−), in contrast to type I NBs where both proteins are present (Dpn+ Ase+). NB II then generate INPs which pass through a maturation process (from Dpn− Ase− to Dpn− Ase+ to Dpn+ Ase+), before producing ganglion mother cells. As reported previously, Brat knockdown causes a massive expansion of NB II-like cells (Dpn+ Ase−) at the expense of INPs (Bowman et al, 2008; Xiao et al, 2012; Janssens et al, 2014; Komori et al, 2018). Brain hemispheres were enlarged, with the dorsal part being nearly completely covered with Dpn+ Ase− cells without signs of further lineage progression (Fig 3D). Spt5-knockdown resulted in a strong suppression of the overgrowth phenotype caused by Brat knockdown and reduced the total number of cells within each lineage (Fig 3D). Distinct GFP-positive cell clusters were visible similar to the control situation. However, within each cluster, most cells still displayed NB II characteristics (Dpn+ Ase−) and only few cells expressed Ase as a marker for INP maturation (Fig 3D). Based on these observations we concluded that, although Spt5-knockdown cannot efficiently revert transformed NB II-like cells

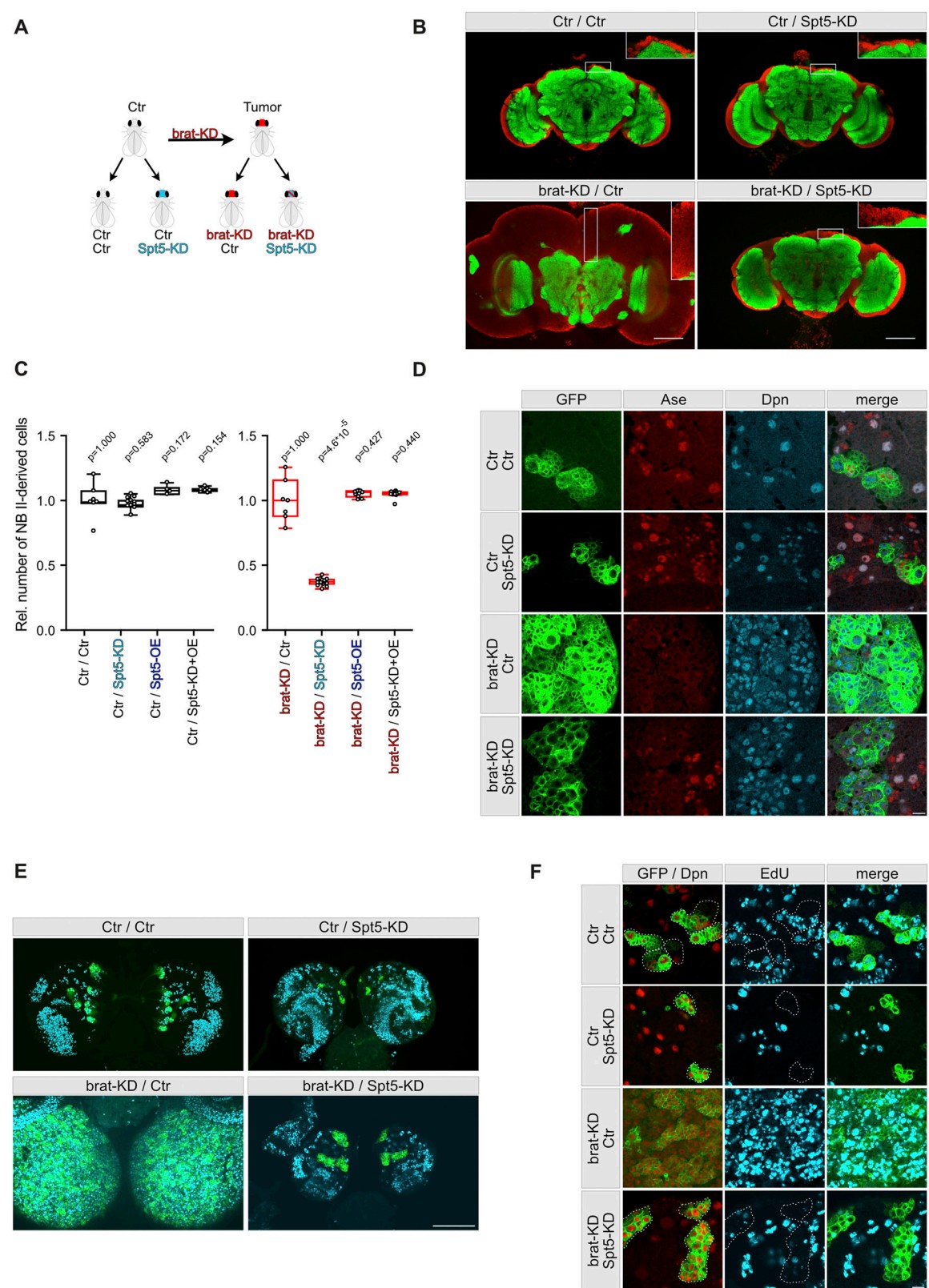

**Figure 3. Spt5 knockdown reduces growth of Brat-depleted tumors.**
**(A)** Scheme of the NB II tumor model, showing expression of luciferase and/or Brat-dsRNA and/or Spt5-siRNA in NB II. **(B)** Adult brains from 5–6-d-old animals were stained for the synaptic protein Bruchpilot (green) to label neuropil structures and the nuclear membrane protein Lamin (red) to visualize the brain cortex. Single pictures were taken at the level of the ellipsoid body of the central complex. Scale bar: 50 μm. **(C)** Number of NB II-derived cells relative to that of control flies (as determined by

into further differentiated INPs, it nevertheless has a major negative impact on tumor formation, possibly by interfering with NB II proliferation. We confirmed this assumption by pulse labeling S-phase cells with EdU in larval brains (Fig 3E and F). Knockdown of Spt5 alone or in combination with brat strongly reduced EdU-incorporation within the GFP-labeled cell clones (highlighted areas; note that most remaining EdU-positive cells do not express GFP and hence are not derived from NB II), whereas the brains with Brat knockdown alone contained multiple cells in S-phase that actively incorporated EdU. Although we noticed a moderate increase in apoptotic cells (positive for the cleaved effector caspase Dcp-1) in Brat /Spt5 knockdown conditions within the GFP-labeled cell clones (Fig S3B), the major tumor suppressive mechanism of Spt5-knockdown can be ascribed to impaired proliferation.

### Effects of Spt5 on tumor transcriptomes

To identify the molecular basis of the observations described above, we isolated NB II from 96 h-old larvae and analyzed their transcriptomes. As shown in Fig 4A, control and Brat-/Spt5-co-depleted cells were highly similar to each other and clearly distinct from Brat-depleted (tumorous) cells with respect to principal component 1, consistent with the reversion of overgrowth by Spt5 knockdown.

Comparison of control with Brat-depleted neuroblasts revealed several alterations of uncharacterized genes (shown in grey) and expected changes in gene expression (Fig 4B and C): brat levels were clearly reduced, whereas the IGF-II mRNA-binding protein Imp (Samuels et al, 2020), the long noncoding RNA cherub (Landskron et al, 2018), the mitochondrial fusion factor Marf (Bonnay et al, 2020), Myc, and Myc target genes (Betschinger et al, 2006; Neumuller et al, 2013; Herter et al, 2015), and glycolytic enzymes (van den Ameele & Brand, 2019; Bonnay et al, 2020) were all strongly upregulated in response to Brat knockdown. All of these changes had been observed before and they contribute to the tumorous phenotype. In addition, the transcription factor Foxo and its target Thor/4E-BP were overexpressed in Brat-depleted NB II.

Next, we analyzed the impact of Spt5 knockdown on tumors caused by brat knockdown. Brat levels themselves were not altered, but Myc targets were significantly down-regulated, in line with observations in mammalian cancer cells (Fig 4B and D). The other described genes were moderately (Marf, Imp) or strongly (lncRNA: cherub) reduced in their expression upon Spt5-knockdown (Fig 4D). In addition, Gart (the second enzyme of the purine biosynthesis pathway; Welin et al, 2010) was significantly repressed, and Gadd45 (an inhibitor of cell cycle progression and inducer of apoptosis;

Tamura et al, 2012) was strongly activated. We also noted that the levels of Foxo and Thor/4E-BP dropped in Spt5-knockdown cells. Together, these expression changes are sufficient to explain the reduction in tumor growth and cellular proliferation and most of them can be ascribed to an impairment of Myc-dependent gene activation upon Spt5-knockdown. However, some of the affected genes are not bona fide Myc targets, for example, lncRNA:cherub (Herter et al, 2015). To find other candidate upstream regulators of these genes, we explored publicly available NB II transcriptome data, and found a significant correlation between Spt5-controlled genes and Mediator target genes. Notably, Gart, lncRNA:cherub, Foxo, and Thor all require Mediator for their full expression (Fig 4E; Homem et al, 2014), raising the possibility that Spt5 might affect their expression via an interaction with Mediator.

### Organismal consequences of Spt5 depletion

Despite the massive brain overgrowth upon brat knockdown in NB II lineages, the tumor-bearing animals reached adulthood at expected frequencies (Fig S4A). However, all of them died within less than 10 d of eclosion, whereas most of the control flies were still alive after 60 d (Fig 5A; for statistical significance of various comparisons see Table S1). Myc-knockdown slightly extended the survival of tumor-bearing flies, showing that these tumors are Myc-dependent (Fig S4B); this is consistent with the published reduction of NB II tumor mass by Myc-knockdown (Neumuller et al, 2013; Herter et al, 2015). This survival benefit is presumably limited by a requirement for Myc in NB II, as seen by the reduced longevity upon single Myc-knockdown (Fig S4B). In contrast, Spt5-knockdown did not impair the survival of control flies, but extended the life span of tumor-bearing animals to more than 26 d (Fig 5A). This rescue was fully reverted by co-expression of an siRNA-resistant version of Spt5, ruling out off-target effects. Overexpression of Spt5 on its own had the opposite effect of Spt5 depletion and significantly shortened the life span of tumorous animals, but had only minor effects on healthy controls. Together, these observations emphasize the importance of Spt5 for abnormal, tumorous tissue growth.

Nevertheless, Brat-/Spt5-knockdown animals did not live as long as control flies, apparently because their tumors recur (Fig 6). In control animals, all brain neuroblasts cease proliferation during metamorphosis and terminally differentiate or undergo apoptosis, such that no more EdU incorporation is detected in adult brains (Fig 6A, compare with Brat-knockdown brains in Fig 6B). Brains from 3-d-old adult Brat-/Spt5-knockdown animals contain clusters of proliferating (EdU-incorporating) cells (Fig 6C); these cells either derive from the few remaining larval EdU-positive cells which never

---

quantification of luciferase activity; see the Materials and Methods section); n = 6–16 single adult flies per genotype. **(D)** NB II lineages in brains from third instar larvae were marked with mCD8::GFP (green) and co-stained for the nuclear proteins Dpn and Ase to distinguish the large NB II (Dpn+ Ase−), newborn intermediate neural progenitors (INPs) (Dpn− Ase−), immature INPs (Dpn− Ase+), and mature INPs (Dpn+ Ase+). Neighboring type I NBs co-express Dpn and Ase. In control brains, two out of eight NB II lineages per brain hemisphere are shown. Spt5 knockdown causes incomplete NB II lineages, whereas Brat knockdown results in massive expansion of cells with characteristics of NB II (Dpn+ Ase−). In the double-knockdown, separate clusters like those in controls are observed, but cells maintained mostly NB II characteristics and only few cells expressed Ase as an indicator of further differentiation. Scale bar: 10 μm. **(E)** EdU incorporation (cyan) in S-phase cells within a period of 90 min in whole-mount brain preparations. Compact EdU signals are seen in the lateral regions representing the proliferation centers of the optic lobes, dispersed signals are evident in the central brain with NB II and their lineages labeled in green. Scale bar: 100 μm. **(F)** In higher magnifications, many proliferating cells outside and within NB II lineages (outlined with dashed lines) are seen in controls, with a strong increase upon brat-KD. No EdU-positive cells are detected in NB II lineages under Spt5-KD and Brat-KD/Spt5-KD conditions. Scale bar: 10 μm.

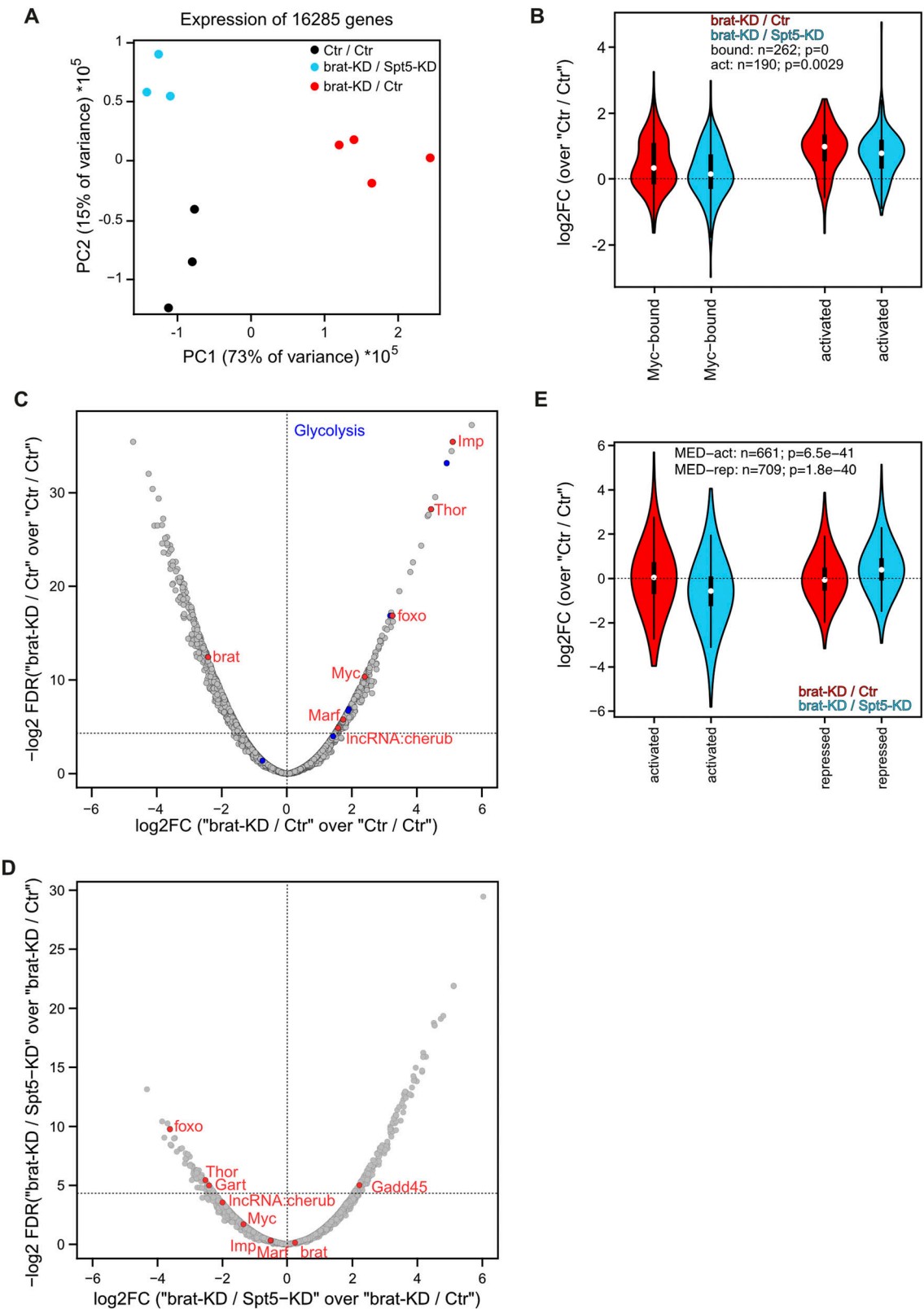

**Figure 4. Effects of Brat and Spt5 knockdown on NB II transcriptomes.**
**(A)** Principal component analysis of NB II transcriptomes from control (black, "Ctr/Ctr"), tumorous (red, "brat-KD/Ctr") or tumorous flies with Spt5 knockdown (blue, "brat-KD/Spt5-KD"). **(B)** Expression levels of Myc-bound or -activated genes that were previously identified in cultured S2 cells (Herter et al, 2015) in Brat-depleted NB II "brat-KD/Ctr" relative to control NB II "Ctr/Ctr" (red), and in Spt5-/Brat-co-depleted NB II "brat-KD/Spt5-KD" relative to control NB II "Ctr/Ctr" (blue). **(C, D)** Volcano plots

stopped proliferating (Fig 3F), or from larval EdU-negative cells which have reentered the cell cycle during metamorphosis. These EdU-positive cells expand over time to ultimately cover large areas and presumably kill the animals (Fig 6C–E). We also noticed very small GFP+/EdU+ cell clusters in young adults, which might indicate infiltration of tumor cells into healthy brain tissue (arrows in Fig 6C and D).

This tumor relapse also raises questions as to the cellular composition of the original tumor tissue. Previous studies demonstrated heterogeneity in different *Drosophila* brain tumor models (Narbonne-Reveau et al, 2016; Genovese et al, 2019), with cells expressing both Imp and the transcription factor Chinmo being the major drivers for tumorigenesis. Both proteins were absent from control NB II lineages (Fig S5A), whereas Brat-knockdown larval (Fig S5B) and adult brains (Fig S5D) contained clones of cells that are either positive for Chinmo and Imp or negative for both, reflecting the described cellular heterogeneity (Genovese et al, 2019). In contrast, in larval Brat-/Spt5-knockdown brains most cells in the NB II-like clusters expressed at least some Chinmo and Imp (Fig S5C); the few exceptions might correspond to mitotic or more differentiated cells. In adult Brat-/Spt5-knockdown brains, large cell clones were heterogenous with respect to Chinmo and Imp expression (Fig S5E), whereas small cell clones only contained Chinmo- and Imp-expressing cells (Fig S5F). These data suggest that Spt5-knockdown has a major impact on tumor heterogeneity in larval brains, by suppressing either the generation or the proliferation of Chinmo/Imp-negative cells. Although impaired in their proliferation at larval stage, (some of) the Chinmo/Imp-expressing cells resume proliferation after adult eclosion and develop cellular heterogeneity.

### Effect of systemic Spt5 knockdown

To explore whether this dependency could potentially be exploited in a curative context, we modified the NB II tumor model. In this new setup, NB II tumors are induced with the same Brat knockdown transgene as used above. In contrast, Spt5 knockdown is driven by the Actin5C promoter that is ubiquitously active in the entire organism. This transgene is initially activated by a heat shock, administered to 120-h-old larvae (well after the onset of GAL4-expression driving Brat knockdown in NB II; Albertson et al, 2004) and remains active thereafter (Fig 5B). The transgene expresses the same Spt5-siRNA as used in the earlier setup, although at a lower level, because this approach does not involve any GAL4/UAS amplification loop (when assayed in whole larval extracts in an analogous qRT-PCR as above for the UAS-siSpt5 transgene, this transgene reduced Spt5 transcript levels only to 88% ± 24%). When flies carrying Brat- and Spt5-knockdown transgenes were reared in the absence of heat shock, they succumbed to tumors within 10 d of adult eclosion; control flies lacking the Brat knockdown transgene had the expected life span (Fig 5C). After heat shock, flies lacking the

siSpt5 transgene showed an analogous behavior. However, in combination with heat shock, the siSpt5 transgene almost doubled the lifespan of tumorous flies (Fig 5C; Table S1). We conclude that systemic targeting of Spt5 is beneficial for cancer-bearing flies.

## Discussion

Several experimental approaches allow the identification of potential cancer drug targets at a medium- to large-scale level. These include the analysis of gain-of-function or overexpression mutations in human tumor samples, systematic knockdown or knockout screens in human cancer cell lines (e.g., Boehm & Golub, 2015), silencing or depletion of candidate genes in mouse transplant models. However, targets identified by these approaches could also be relevant for healthy tissues. It is therefore essential to determine the "therapeutic window" of any putative target. This is usually done by analysing appropriate mouse models, containing, for example, floxed target genes in combination with an OHT-activatable Cre recombinase, or expressing shRNAs against the target gene. Such approaches are more laborious and expensive than the initial genetic screens, and hence therapeutic windows are often addressed only once target-specific inhibitors are available, resulting in high attrition rates at late pre-clinical stages. Our present analysis suggests that *Drosophila* can be used to reveal the existence of such therapeutic windows.

The elongation factor Spt5 initially caught our attention because of its physical interaction with Myc in cultured human cancer cells (Baluapuri et al, 2019). Here, we found that it also collaborates with Myc functionally in vivo. Simultaneous reduction of both proteins during larval development synergistically impaired the growth of imaginal tissue, consistent with the notion that Myc-dependent efficient activation of growth-promoting genes requires association with Spt5. Combining Spt5-knockdown with Myc-overexpression during post-proliferative eye disc development resulted in a striking novel phenotype, indicative of massive apoptosis not seen with either treatment alone. This could indicate that some Myc targets do not require Spt5 for their expression, and that the balance of Spt5-dependent and -independent targets determines the biological outcome of Myc activation, for example, tissue growth versus attrition (similar to what was suggested by Steiger et al [2008]). Alternatively, combined Spt5 knockdown and Myc overexpression might titrate Spt5 away from some genes, affecting their expression and resulting in the observed phenotype (similar to what was suggested by Baluapuri et al [2019]).

We used Spt5 as an example of an essential Myc co-factor and evaluated the consequences of knocking down Spt5 in a Myc-dependent NB II brain tumor model. In a first approach, we used the same expression system to target both Brat (to generate the NB II tumors) and Spt5 specifically in NB II. In this setting, Spt5-knockdown almost completely reverted the tumorous tissue

showing expression in Brat-depleted NB II (tumors) relative to control NB II (C), and in Spt5-/Brat-co-depleted NB II relative to Brat-depleted NB II (D). Horizontal lines mark significance level (FDR Q-value) of 0.05. Labeled genes are described in the text; for a complete listing of all genes, see Tables S2 and S3. **(E)** Expression levels of previously identified Med27-activated or -repressed genes in Brat-depleted NB II relative to control NB II (red), and in Spt5-/Brat-co-depleted NB II relative to control NB II (blue). *P*-values are derived from a paired *t* test.

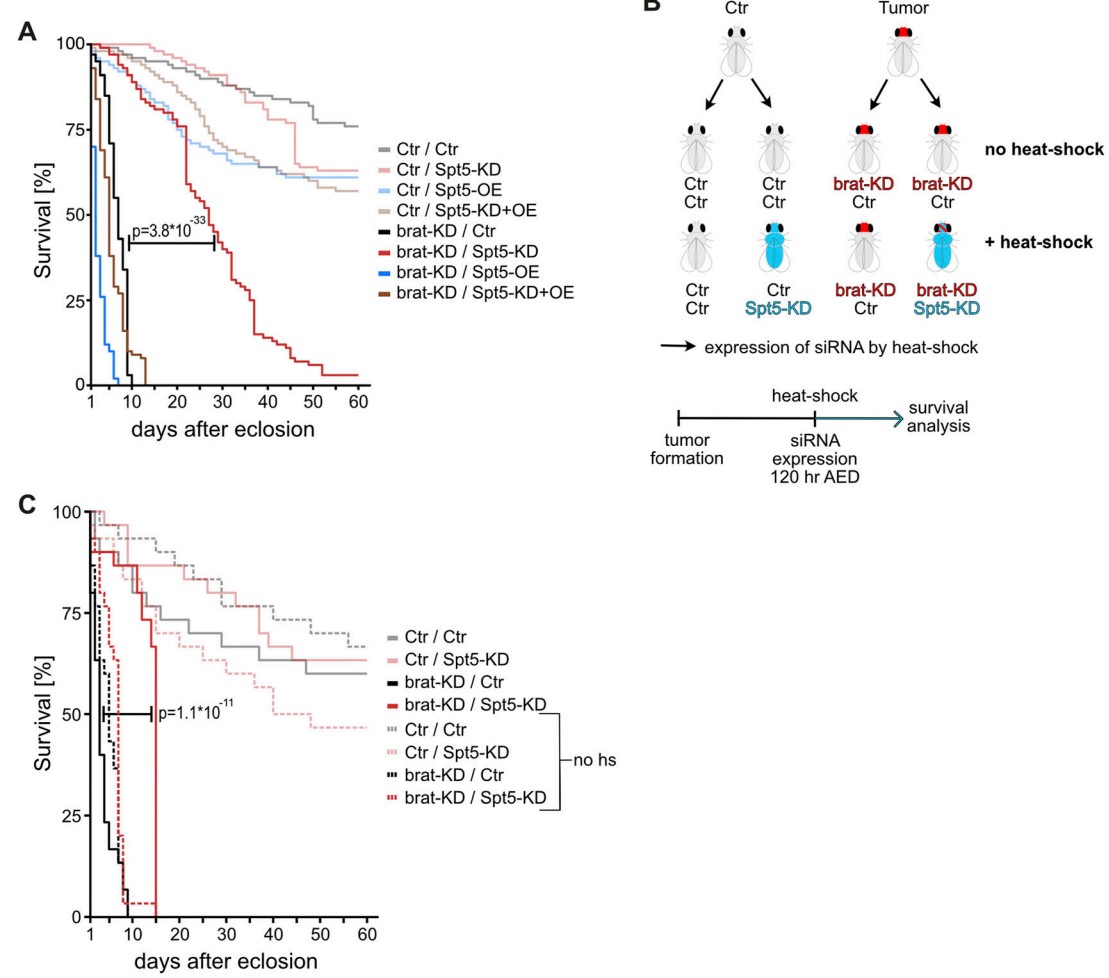

**Figure 5. Impact of Spt5 knockdown on longevity of tumorous flies.**
**(A)** Survival of male flies with the indicated genotypes in days after adult eclosion (n = 100 flies for each genotype). **(B)** Scheme for ubiquitous and temporally controlled Spt5 depletion in tumorous and control animals (for details, see text). **(C)** Survival of male flies with the indicated genotypes +/– heat shock induced ubiquitous Spt5 depletion days after adult eclosion. Spt5 knockdown significantly extended the lifespan of tumorous flies ($P = 1.1 \times 10^{-11}$; n = 30 flies for each genotype).

overgrowth in larval brains and more than tripled adult animal survival. However, most of the larval NB II-derived cells retained neuroblast characteristics and expressed Imp and Chinmo. These cells apparently did not finally exit the cell cycle and terminally differentiate during metamorphosis, but instead regained a high proliferative potential after adult eclosion. We do not know whether these relapsing tumor cells somehow recovered their Spt5 expression or whether they proliferated despite continually low levels of Spt5.

It was previously shown that Chinmo/Imp cells are the major drivers for NB II tumorigenesis; they propagate by repeated self-renewing divisions, but at low frequencies also spawn Chinmo/Imp-negative cells which have a reduced proliferative capacity (Genovese et al, 2019). In line with these findings, we found large tumor clones in adult Brat-/Spt5-knockdown brains to be heterogenous with respect to their Imp/Chinmo expression status (Fig 6E), whereas all cells in small clones expressed both Chinmo and Imp and also incorporated EdU (Figs 6C and D and S5F). These

observations could suggest that Chinmo/Imp-expressing cells have invasive properties and generate metastases in healthy tissue.

Importantly though, knockdown of Spt5 in selected neuroblasts of control animals without brain tumors had mild effects on brain development, and did not negatively impact adult survival, demonstrating the potential value of Spt5 as a therapeutic target. However, in clinical settings, it is typically not possible to direct a therapy exclusively at transformed cells and therapeutic intervention cannot be initiated at early tumor development. For this reason, we developed a second system that allowed temporal separation of tumor initiation and Spt5 knockdown, and that targeted Spt5 not only in NB II but throughout the organism. Whereas this approach relied on the same system to deplete Brat and the same Spt5-targeting siRNA as the first approach above, the latter was induced by a heat shock and directly driven by the Actin5C promoter rather than being amplified by a GAL4/UAS loop, resulting in lower siRNA expression and less efficient Spt5 depletion in NB II. Nevertheless, Spt5 knockdown had a strong therapeutic benefit for

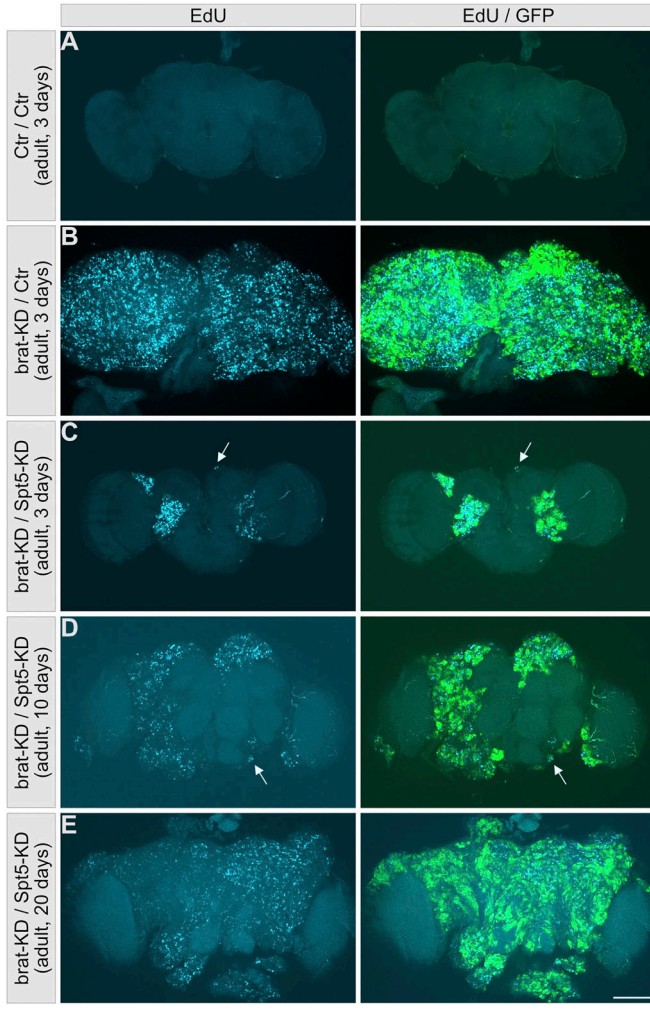

**Figure 6. Tumor relapse in Brat/Spt5-knockdown brains.**
**(A, B, C, D, E)** EdU incorporation (cyan) in S-phase cells within a labeling period of 90 min in whole-mount brain preparations from 3, 10, and 20 d old flies of the indicated genotypes. NB II lineages are marked with mCD8::GFP (green). **(A)** Because neuroblasts stop proliferation during metamorphosis, neither NB II cell lineages nor EdU incorporation are detected in control brains. **(B)** Upon Brat knockdown, proliferating neuroblasts cover the whole brain. **(C, D, E)** A progressive relapse of proliferating cells and increase in cell cluster size is evident in Brat-/Spt5-knockdown brains ((C): 3 d; (D): 10 d; (E): 20 d). In addition, very small, EdU-positive cell clusters are observed (arrows). Scale bar: 100 µm.

tumorous flies, as it almost doubled their survival time. Importantly, ubiquitous Spt5-knockdown did not impair the survival of tumor-free control animals, nor did heat-shocks per se have any deleterious effect on longevity. A molecular explanation for this tumor-suppressive effect is provided by our analysis of NB II transcriptomes: Spt5-knockdown resulted in strong down-regulation of several genes associated with NB II transformation, and an up-regulation of genes opposing uncontrolled proliferation. Most of these expression changes can be ascribed to a reduction of Myc:Spt5 complexes, whereas some probably reflect a functional interaction of Spt5 with the Mediator complex, which itself plays a role in NB II tumor formation. As expected, Myc knockdown also extended the longevity of tumor-bearing flies, but this effect

was less pronounced than for Spt5 knockdown. This difference might indicate that Myc:Spt5 complexes are more critical for transformed cells than for normal tissues. In any case, our experiments demonstrate that targeting a protein, Spt5, which was selected based on its physical interaction with Myc, can reduce tumor mass and provide a survival benefit for tumor-bearing animals, even though this protein is essential for normal development. It is open, though, whether Spt5 is the best-suited target in Myc-dependent cancers, as many additional proteins have been shown to bind Myc. Our *Drosophila*-based approach allows a simple pre-screening of these candidates to filter for the best targets that can subsequently be funneled into more laborious analyses in mice.

# Materials and Methods

### Flies

Sources of flies: "GMR-GAL4" and "GMR-GAL4 3x(UAS-Myc)" were characterized by Secombe et al (2007), Montero et al (2008), and Steiger et al (2008); "wor-GAL4 ase-GAL80 UAS-GFP UAS-Luciferase" and "wor-GAL4 ase-GAL80 UAS-brat-IR UAS-GFP UAS-Luciferase" were initially generated by Neumuller et al (2013) and also described in Herter et al (2015); "ey-FLP tub-FRT-Myc-FRT-GAL4" was described in Bellosta et al (2005); "act-FRT-stop-FRT-GAL4" was, for example, used in Gerlach et al (2017); UAS-Spt5 (resistant to siSpt5) (Qiu & Gilmour, 2017); UAS-siSpt5 (Bloomington stock number B-34837; Perkins et al, 2015); mutant allele "Spt5[SIE-27]" (Mahoney et al, 2006). "act-FRT-stop-FRT-siSpt5" was generated by inserting "AggccagtCAGAAGCTACAGTCCATTCAAtagttatattcaagca-taTTGAATGGACTGTAGCTTCTGgcggccAGTC" ("siSpt5_f") into pAct-FRT-stop-FRT3-FRT-FRT3-GAL4_attB (#52889; Bosch et al, 2015; AddGene vector). The resulting construct pACT5C-FRT-stop-FRT-siSpt5 was inserted in ZH86Fb by GenetiVision Corp.

### Genetic manipulation of Spt5 and Myc in eyes

Eye-specific reduction of Myc levels as used for Fig 2C and D was described by Bellosta et al (2005). Briefly, Myc cDNA was ubiquitously expressed under the control of the tubulin promoter by the transgene "tub-FRT-Myc-FRT-GAL4" (inserted on the X-chromosome), which increases Myc levels to <180% as compared with control (Wu & Johnston, 2010). The same X-chromosome carries an eyFLP transgene, which eliminates the Myc cDNA specifically in eye imaginal disc cells, resulting in expression of GAL4 instead. Flies designated as "Myc[P0]" additionally carry the hypomorphic allele *Myc[P0]* on the same X chromosome, whereas "ctr" flies are WT for Myc and only carry the two described transgenes. Hence, eye imaginal discs of the *Myc[P0]* flies described in Fig 1C and D are mutant for Myc specifically in the eye primordia, thus expressing less than 40% of Myc mRNA. Importantly, this *Myc* allele only reduces the amount of Myc protein, but does not alter its amino acid sequence.

## Targeted expression

Type II neuroblasts were targeted by a combination of worniu (wor)-GAL4, which is expressed in type I and II NBs, and asense (ase)-GAL80 to repress GAL4 activity in the type I NBs (Neumüller et al, 2011).

To knock down Spt5 ubiquitously after the onset of tumor generation, the system above (wor-GAL4 ase-GAL80 UAS-Brat-KD) was combined with the transgenes "hs-FLP" and "pACT5C-FRT-stop-FRT-siSpt5." siSpt5 expression was initiated by transferring larvae at 120 h after egg deposition for 1 h to a water bath at 37°C.

## Confocal microscopy

For immunostainings, brains from late third instar larvae or adults were dissected in PBS (10 mM $Na_2HPO_4$, 2 mM $KH_2PO_4$, 2.7 mM KCl, 137 mM NaCl) and fixed on ice for 25 min in PLP solution (4% PFA, 10 mM $NaIO_4$, 75 mM lysine, 30 mM sodium phosphate buffer, pH 6.8). All washings were done in PBT (PBS plus 0.3% Triton X-100). After blocking in PBT containing 5% normal goat serum for 1 h, tissues were incubated overnight at 4°C with combinations of the following primary antibodies: rabbit anti-Ase (1:400; F. Diaz-Benjumea), mouse anti-Bruchpilot (1:30, clone nc82; E Buchner), rat anti-Chinmo (1:500; N Sokol), rabbit anti-Dcp-1 (1:100, # 9578; Cell Signaling Technology), guinea pig anti-Dpn (1:1,000; J Knoblich), chicken anti-GFP (1:1,500 #ab13970; Abcam), rabbit anti-Imp (1:1,500; F Besse), guinea pig anti-Lamin DmO (1:300; G Krohne). Secondary antibodies conjugated with AlexaFluor 488, Cy3 or Cy5-conjugated were purchased from Molecular Probes and Dianova.

For 5-ethynyl-2'-deoxyuridine (EdU) labeling, brains from third instar larvae or adults were dissected in PBS and incubated with 20 $\mu$M EdU in PBS for 90 min. After fixation in 4% PFA for 15 min, followed by immunostaining, before EdU incorporation into replicating DNA was detected with the Click-iT Alexa Fluor 647 EdU imaging kit (Thermo Fisher Scientific [Invitrogen]).

Embedding of brains was done in Vectashield (Vector Laboratories) and confocal images were collected with a Leica SPE or SP8 microscope (Leica Microsystems). Image processing was carried out with the ImageJ distribution Fiji (Schindelin et al, 2012).

## Phenotypic analysis

To measure adult eye sizes, adult males were collected at 1–7 d after eclosion and killed by freezing. One eye per individual fly was photographed on a Zeiss Discovery.V8 stereomicroscope fitted with a 1.5x lens and processed with Axiovision Extended Focus software and the ImageJ distribution Fiji.

To measure luciferase activity, male flies were collected within 1 d of adult eclosion and frozen individually at −20°C until use. Each fly was then lysed in 50 $\mu$l Passive Lysis Buffer (Promega) and homogenized with ~10 steel beads in a "Bullet Blender Blue" Homogenizer at speed 10 for 2 min, followed by a 4' centrifugation at 12,000$g$. 10 $\mu$l of the supernatant was transferred into a black 96-well plate and assayed for luciferase expression using the Dual Luciferase Reporter Assay System in an automated luminometer. Note that the tumorous brat–KD flies contain a "UAS-Fire-flyLuciferase" transgene, whereas the non-tumorous flies without

the brat–KD carry a "UAS-RenillaLuciferase" transgene (see Neumuller et al [2013]; Herter et al [2015]). Hence, luciferase activities reflect the number of GAL4-expressing cells, but they can only be compared within each series of genotypes, not between the brat–WT genotypes (shown in black in Fig 3C) and the brat–KD genotypes (shown in red in Fig 3C).

For weighing flies, 1–4 d old adult flies were dried for 20' at 95° (first for 10' with a closed, then with an opened lid) and then stored at RT. Before weighing on a Mettler UMT5 Comparator scale (Mettler Toledo), the flies were allowed to equilibrate with ambient atmosphere for at least 30'.

To determine duration of development, timed egg lays (5–14 h) were performed and eclosion was monitored two to three times a day.

## Survival analysis

Parents were transferred to a fresh food vial every 3 d. Offspring was collected within 1 d of adult eclosion, and subsequently transferred to fresh vials every 2 d. The number of living flies was monitored daily for a period of 60 d.

## Isolation of type II neuroblasts

Processing of larvae for next-generation sequencing was carried out as described by Harzer et al (2013). Briefly, 5-d old larvae were washed sequentially in PBS, 70% ethanol, and Schneider's medium. Within ≤1 h larvae were dissected and brains transferred to a 0.5 ml low-binding Eppendorf tube containing Rinaldini's solution (8 g/liter NaCl, 0.2 g/liter KCl, 50 mg $NaH_2PO_4$, 1 g/liter $NaHCO_3$, 1 g/liter glucose). After two washes, Rinaldini's solution was replaced with a dissociation solution (Schneider's medium containing 100 ml/liter heat-inactivated fetal bovine serum, 2 ml/liter insulin, 20 ml/liter penicillin–streptomycin, 100 ml L-glutamine, 20 mg/liter L-gluta-thione, 20 mg/ml collagenase I, 20 mg/ml papain), and the brains were stirred up by pipetting. After 1 h incubation at 30°C with occasional mixing, the brains were washed twice with Rinaldini's solution and with Schneider's medium, and then mechanically dissociated by pipetting. The resulting cell suspension was filtered through a 30-$\mu$m mesh 5-ml FACS tube, which was then filled up with Schneider's medium to a total volume of 10 $\mu$l per dissected larval brain and sorted in a BD FACSAria III sorter. Type II Neuroblasts were identified based on side scatter (SSC), forward scatter (FSC) and GFP intensity, collected into 96-well microtiter plates, containing 1 $\mu$l $\beta$-mercaptoethanol and 100 $\mu$l Lysis Buffer (Agilent Technologies Absolutely RNA Nanoprep Kit) per well, and subsequently stored at −80°C until use.

## qRT–PCR

To induce expression of siSpt5 or of an Spt5-cDNA, flies of the genotype "hs-FLP act-FRT-stop-FRT-GAL4 UAS-siSpt5," "hs-FLP act-FRT-stop-FRT-GAL4 UAS-Spt5" or "hs-FLP act-FRT-stop-FRT-siSpt5" were exposed to a 1-h heat shock at 37°. 1 d later, single wandering larvae were processed for total RNA extraction, reverse transcription, and qRT-PCR as described in Montero et al (2008), using the following primers: alphaTub84B-fwd

5'-GCCAGATGCCGTCTGACAA-3', alphaTub84B-rev2 5'-AGTCTCGCTGAA-GAAGGTGTTGA-3'; Spt5-fwd2 5'-GCTCTCAATCGGGCCACT-3', Spt5-rev2 5'-GGATTCATCGCTCTTGCCG-3'; Spt5-fwd3 5'-TGCAAAACGCCACTTTGGAG-3', Spt5-rev3 5'-GCCGGGCAATAGAGTTTGTTG-3'.

## mRNA library preparation

RNA was isolated using Agilent Technologies' Absolutely RNA Nanoprep Kit (including DNase I digestion). RNA concentration and quality were determined on 2100 Bioanalyzer Instrument (Agilent Technologies) using the Agilent RNA 6000 Pico Kit (Agilent Technologies). Library preparation was performed using the Poly(A) mRNA Magnetic Isolation Module (New England Biolabs) and the NEBNext Ultra II Directional RNA library Prep Kit for Illumina (New England Biolabs). For library amplification, 17 or 24 PCR cycles were used. Library size distribution and concentration were analyzed on the Fragment Analyzer (Agilent Technologies) using the NGS Fragment High Sensitivity Analysis Kit (1–6,000 bp; Agilent Technologies). The libraries were sequenced on Illumina instrument (NEXTSeq500).

## Bioinformatics

Bliss synergy scores (Bliss, 1939) were calculated using the R package synergyfinder 1.10.7 (Zheng et al, 2022), where scores >10 suggest a synergistic interaction; n = 6–10 collections per genotype for Fig 2A, median derived of eight flies for each genotype for Fig 2C and D.

For RNAseq analysis, reads were mapped to version BDGP6 of the *Drosophila* genome, using bowtie2 with the setting "very-sensitive-local" (Langmead & Salzberg, 2012) (2.2–9.8 million mapped reads per sample). Differentially expressed genes were identified using edgeR 3.26.8 (Robinson et al, 2010). Gene set enrichment analysis was carried out with GSEA 4.0.2. (Subramanian et al, 2005) and GO terms obtained from the ENSEMBL annotation for BDGP6.32. Volcano & box plots were generated in R.

## Relevant genotypes

### Figs 1C and D and S1
GMR-GAL4/+
GMR-GAL4/+; UAS-siSpt5/+
GMR-GAL4/+; UAS-Spt5/+
GMR-GAL4/+; UAS-Spt5 UAS-siSpt5/+
GMR-GAL4 3x(UAS-Myc)/+
GMR-GAL4 3x(UAS-Myc)/+; UAS-siSpt5/+
GMR-GAL4 3x(UAS-Myc)/+; UAS-Spt5/+
GMR-GAL4 3x(UAS-Myc)/+; UAS-Spt5 UAS-siSpt5/+

### Figs 2A and S2A
+/Y
+/Y; Spt5[SIE-27]/+
dm[P0]/Y
dm[P0]/Y; Spt5[SIE-27]/+

### Figs 2C and D and S2B–I
tub-FRT-Myc-FRT-GAL4 ey-FLP/Y

tub-FRT-Myc-FRT-GAL4 ey-FLP/Y; UAS-siSpt5/+
tub-FRT-Myc-FRT-GAL4 ey-FLP/Y; UAS-Spt5/+
tub-FRT-Myc-FRT-GAL4 ey-FLP/Y; UAS-Spt5 UAS-siSpt5/+
dm[P0] tub-FRT-Myc-FRT-GAL4 ey-FLP/Y
dm[P0] tub-FRT-Myc-FRT-GAL4 ey-FLP/Y; UAS-siSpt5/+
dm[P0] tub-FRT-Myc-FRT-GAL4 ey-FLP/Y; UAS-Spt5/+
dm[P0] tub-FRT-Myc-FRT-GAL4 ey-FLP/Y; UAS-Spt5 UAS-siSpt5/+

### Figs 3B, D, E, 4, and 6, S3, S4A, and S5
wor-GAL4 ase-GAL80 UAS-mCD8::GFP
wor-GAL4 ase-GAL80 UAS-mCD8::GFP UAS-Brat-KD
wor-GAL4 ase-GAL80 UAS-mCD8::GFP UAS-siSpt5
wor-GAL4 ase-GAL80 UAS-mCD8::GFP UAS-Brat-KD UAS-siSpt5

### Fig 3C
wor-GAL4 ase-GAL80 UAS-RLuc
wor-GAL4 ase-GAL80 UAS-RLuc UAS-siSpt5
wor-GAL4 ase-GAL80 UAS-RLuc UAS-Spt5
wor-GAL4 ase-GAL80 UAS-RLuc UAS-siSpt5 UAS-Spt5
wor-GAL4 ase-GAL80 UAS-FLuc UAS-Brat-KD
wor-GAL4 ase-GAL80 UAS-FLuc UAS-Brat-KD UAS-siSpt5
wor-GAL4 ase-GAL80 UAS-FLuc UAS-Brat-KD UAS-Spt5
wor-GAL4 ase-GAL80 UAS-FLuc UAS-Brat-KD UAS-siSpt5 UAS-Spt5

### Fig 5A
wor-GAL4 ase-GAL80 UAS-RLuc
wor-GAL4 ase-GAL80 UAS-RLuc UAS-siSpt5
wor-GAL4 ase-GAL80 UAS-RLuc UAS-Spt5
wor-GAL4 ase-GAL80 UAS-RLuc UAS-siSpt5 UAS-Spt5

### Fig 5C
hs-FLP wor-GAL4 ase-GAL80
hs-FLP wor-GAL4 ase-GAL80 UAS-Brat-KD
hs-FLP wor-GAL4 ase-GAL80 act-FRT-stop-FRT-siSpt5
hs-FLP wor-GAL4 ase-GAL80 UAS-Brat-KD act-FRT-stop-FRT-siSpt5

### Fig S4B
wor-GAL4 ase-GAL80
wor-GAL4 ase-GAL80 UAS-Brat-KD
wor-GAL4 ase-GAL80 UAS-Myc-KD
wor-GAL4 ase-GAL80 UAS-Brat-KD UAS-Myc-KD

# Data Availability

RNA expression data are available at the Gene Expression Omnibus under the accession number GEO: GSE220110.

# Supplementary Information

# Life Science Alliance

# Acknowledgements

We thank David Gilmour, Florence Besse, Erich Buchner, Fernando Diaz-Benjumea, Jürgen Knoblich, Georg Krohne, Nicholas Sokol, and Lesley Weaver for generously providing antibodies and fly stocks, Hugo Stocker for critical reading of the article, the DFG grants WO 2108/1-1 and GRK 2243 to E Wolf, and the ERC for grant TarMyc to E Wolf.

## Author Contributions

J Hofstetter: conceptualization, data curation, formal analysis, investigation, methodology, and writing—original draft, review, and editing.

A Ogunleye: formal analysis, investigation, and methodology.

A Kutschke: investigation.

LM Buchholz: investigation.

E Wolf: conceptualization, supervision, funding acquisition, project administration, and writing—original draft, review, and editing.

T Raabe: conceptualization, data curation, formal analysis, supervision, funding acquisition, project administration, and writing—original draft, review, and editing.

P Gallant: conceptualization, data curation, formal analysis, supervision, methodology, project administration, and writing—original draft, review, and editing.

## Conflict of Interest Statement

The authors declare that they have no conflict of interest.

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
