## [Reviewer comments · Life Science Alliance]

Life Science Alliance

Spt5 interacts genetically with Myc and is limiting for brain tumor growth in *Drosophila*

Julia Hofstetter, Ayoola Ogunleye, André Kutschke, Lisa Buchholz, Elmar Wolf, Thomas Raabe, and Peter Gallant
DOI: <https://doi.org/10.26508/lsa.202302130>

Corresponding author(s): Peter Gallant, University of Würzburg and Elmar Wolf, University of Würzburg

Review Timeline:

Submission Date:	2023-05-03
Editorial Decision:	2023-06-16
Revision Received:	2023-09-15
Editorial Decision:	2023-10-10
Revision Received:	2023-10-12
Accepted:	2023-10-13

Transaction Report:

June 16, 2023

Re: Life Science Alliance manuscript #LSA-2023-02130-T

Dr. Peter Gallant
University of Würzburg
Department of Biochemistry and Molecular Biology, Theodor Boveri Institute, Biocenter
Am Hubland
Würzburg, Bavaria 97074
Germany

Dear Dr. Gallant,

Thank you for submitting your manuscript entitled "Spt5 interacts genetically with Myc and is limiting for brain tumor growth in *Drosophila*" to Life Science Alliance. The manuscript was assessed by expert reviewers, whose comments are appended to this letter. We invite you to submit a revised manuscript addressing the Reviewer comments.

Thank you for this interesting contribution to Life Science Alliance. We are looking forward to receiving your revised manuscript.

Sincerely,

B. MANUSCRIPT ORGANIZATION AND FORMATTING:

Reviewer #1 (Comments to the Authors (Required)):

The manuscript by Hofstetter et al. describes well-known Drosophila-based approaches to show physical and functional interaction between Myc and Spt5, which are already known in different organisms in the context of brain tumors. The authors show that Spt5 downregulation can reduce Myc-dependent tumor mass and provide a survival benefit for tumor-bearing flies while it did not negatively impact healthy cells. Although the Myc-Spt5 interaction in this study is based on phenotypic and genetic data, it provides a potential value of Spt5 as a therapeutic target in myc-driven tumors. As such this paper is of interest to the fields of transcription and cancer biology. However, the study lacks controls and has multiple problematic issues that should be addressed before publication.

Comments:

1. Throughout the paper, the authors use KD and OE techniques of both Spt5 and Myc. It seems logical to show at least once the proteins or mRNA levels (in the case of siRNA) upon KD and OE.
2. Many data are presented without showing individual data points, preventing the assessment of effect size and variability.
3. Figure 1D. It will be much more understood if instead of "myc" (black), it is written "control" as in Figure 1C.
4. Page 3, lines 39-41 "Together, these observations demonstrate that Myc and Spt5 functionally interact and suggest that the transcriptional program activated by excessive Myc levels is critically dependent on Spt5"- if indeed, as stated, Spt5 is essential for MYC-mediated transcriptional activation, one would expect that KD of Spt5 in Myc overexpressing eye will result similar to control. Please explain.
5. Figure 2A. (i) What are the black dots represent? (ii) $n=6-109$ - is it correct?
6. Figure S2A. The similar colors are very confusing. The authors should choose different colors.
7. Figure 3C. It does not mention what exactly the luciferase activity reflects.
8. Page 4, line 32: The authors should add information indicating what Deadpan (Dpn) and Asense (Ase) are.
9. Figures 3E & 3F. I find some contradiction between these two figures concerning Ctr / Spt5-KD. In 3E it seems that Spt5 KD led to higher EdU incorporation, opposite to the results in Figure 3F and of what is stated in the text (p. 5, lines 2-3). Please clarify.
10. Figure 4B. "ctr" should be "tum" (similar to 4D). Otherwise, it is misleading.
11. Figure 5C. See comment no. 3. Especially there is no difference between black and dark gray.

Reviewer #2 (Comments to the Authors (Required)):

A major goal in cancer therapy is to be able to identify drugs that eliminate tumors while exhibiting minor effects on healthy tissues. Genes of the Myc family are among the most commonly up-regulated genes in human cancers, acting as potent oncogenes. The study uses Drosophila to test the hypothesis that targeting a partner of Myc could prevent its oncogenic activity and stop tumor growth while leaving normal cells unaltered.

First the authors demonstrate that during Drosophila development, Myc and Spt5 (a binding partner of Myc in mammals) genetically interact. Then they convincingly show using a Drosophila brain tumor model that the knockdown of Spt5 is sufficient to significantly alter tumor growth. Although Spt5 is important and necessary for development, the authors nicely show that silencing Spt5 systemically from late larval stages can extend the lifespan of tumor bearing flies.

This work is interesting as it provides a proof of concept that Myc partners can be targeted for cancer therapy to efficiently slow down or stop tumor growth while minimizing the impact on the organism.

The experiments are convincing and clearly presented and the manuscript is clearly written.

I provide here a set of questions and comments that could be addressed in new experiments or in the text and may help to clarify some points and link the transcriptomic data to some of the phenotypes.

- One question is whether Spt5-KD tumor cells are eliminated in the adult or remain silent (non-proliferative) as shown in larvae (Fig 3F). This could be tested by doing an anti-Dpn and anti-PH3 staining in bratKD / Spt5-KD adult flies.

- Fig 5A suggests that the rescue is not complete in brat-KD / Spt5-KD animals as they still die significantly earlier than Ctr /

Spt5-KD. Is it because of a "relapse". Do they die because the tumor ultimately continues growing?

- It has been shown that the growth and maintenance of bratKD tumors rely on a specific sub-population of tumor cells, expressing Imp, that act as cancer stem cells (PMID: 27296804; PMID: 31566561). The transcriptomic analysis suggests that Imp is down-regulated in bratKD / Spt5-KD. It could be easily tested if this population is eliminated in this context using immunostaining, explaining why tumors stop growing.
- On the other hand, if Imp+ tumor cells remain in adults but in a dormant state, it could explain a possible relapse later.
- Does Spt5-OE expand the subpopulation of Imp+ cells in tumors, making them growing faster?
- What is the effect of silencing Myc in normal type-II lineages and in bratKD tumors ? Are tumors eliminated ?
- The authors claim that they can systemically target Spt5 using the act-FRT-stop-FRT-siSpt5 transgene. It is unclear how efficient the hs-FLP-mediated recombination is between FRT sites in their assay. Is it possible to assess this using anti-Spt5 antibody? Can the lifespan be further extended if two or three heat-shocks are performed in order to increase the number of cells that have undergone the recombination event ?

We thank both reviewers for their positive evaluation, and for their insightful (and above all: reasonable) comments. Below, we address them point-by-point.

In the manuscript text the relevant changes have been highlighted in red.

Reviewer #1

The manuscript by Hofstetter et al. describes well-known *Drosophila*-based approaches to show physical and functional interaction between Myc and Spt5, which are already known in different organisms in the context of brain tumors. The authors show that Spt5 downregulation can reduce Myc-dependent tumor mass and provide a survival benefit for tumor-bearing flies while it did not negatively impact healthy cells. Although the Myc-Spt5 interaction in this study is based on phenotypic and genetic data, it provides a potential value of Spt5 as a therapeutic target in myc-driven tumors. As such this paper is of interest to the fields of transcription and cancer biology. However, the study lacks controls and has multiple problematic issues that should be addressed before publication.

Comments:

1. Throughout the paper, the authors use KD and OE techniques of both Spt5 and Myc. It seems logical to show at least once the proteins or mRNA levels (in the case of siRNA) upon KD and OE.

To assess variations of Spt5 levels we measured Spt5 mRNA, since we were unable to obtain an antibody against *Drosophila* Spt5. First, based on our RNAseq data (the result of which is graphically shown in Fig 4D) Spt5-KD in NB II reduces Spt5 levels to 49% (ratio of "Spt5-/brat-KD NB II" over "brat-KD NB II").

Second, to independently evaluate our Spt5 transgenes, we induced ubiquitous expression of the Spt5-cDNA or of siSpt5 by a 1-hour heat-shock at 37 °C and analysed mRNA levels 1 day later in wandering larvae; as controls we used the corresponding genotypes without heat-shock. This resulted in the following expression levels:

for "hs-FLP actin5C-FRT-stop-FRT-GAL4 UAS-Spt5" (Spt5 overexpression): 750% (+/- 90%) of control;

for "hs-FLP actin5C-FRT-stop-FRT-GAL4 UAS-siSpt5" (the KD-transgene used for all experiments except Fig 5C): 62% (+/- 24%) of control;

for "hs-FLP actin5C-FRT-stop-FRT-siSpt5" (the KD-transgene used for Fig5C): 88% (+/- 24%) of control.

These values are now added to the text.

For Myc levels we refer to published data. The eye-specific Myc-overexpression system used here in Fig 1C,D was characterized by Secombe et al 2007, *Genes Dev* 21, 537, who analyze larval eye imaginal discs in a Western blot to show a dramatic increase of Myc levels upon overexpression.

[Figure removed by editorial staff per authors' request]

Myc levels in the Myc[P0] mutant were measured by qRT-PCR by Wu & Johnston 2007, *Genetics* 184, 199. Based on their data in Table 1, the samples "MycP0" (Fig 2C) express 20% of the samples "ctr" in developing eyes. Note that all these flies in Fig 2C also carry the transgene "tub-FRT-Myc-FRT-GAL4", but that the Myc-cDNA on this transgene is eliminated throughout the developing eye imaginal discs by the additional transgene "ey-FLP", leading to the expression of "GAL4" instead.

References to these publications are now included in the text.

2. Many data are presented without showing individual data points, preventing the assessment of effect size and variability.

We have amended all box & bar plots (Figures 1D, 2A, 2D, 3C, S3A) to include the individual data points. In addition, the box plots in Figs 4B & 4D (which are derived from a large number of single measurements) were replaced with violin plots.

3. Figure 1D. It will be much more understood if instead of "myc" (black), it is written "control" as in Figure 1C.

In Fig 1D, we have replaced the labels "Myc/Ctr" with "Ctr/Ctr", and "Myc/Spt5-KD" with "Ctr/Spt5-KD", as suggested.

4. Page 3, lines 39-41 "Together, these observations demonstrate that Myc and Spt5 functionally interact and suggest that the transcriptional program activated by excessive Myc levels is critically dependent on Spt5"- if indeed, as stated, Spt5 is essential for MYC-mediated transcriptional activation, one would expect that KD of Spt5 in Myc overexpressing eye will result similar to control. Please explain.

We realize that the term "transcriptional program activated by excessive Myc levels" is ambiguous. We used it to imply the *entirety* of Myc's effects on growth, cell division, apoptosis, differentiation, etc. Upon depletion of Spt5, this entirety is clearly affected, but the individual parts are affected in different ways. It appears that Spt5-depletion abrogates Myc-induced growth, but potentiates Myc-induced apoptosis, resulting in a massively disturbed eye morphology, rather than in a reversion to control phenotype. This apparent increase in Myc-dependent apoptosis could result e.g. from increased expression of apoptosis-inducing genes upon Spt5-knockdown, or from an imbalance between Myc-induced Spt5-dependent and -independent genes. We have expanded the corresponding section in the text to explain more clearly what we mean.

Interestingly, we have previously made a somewhat similar observation when Max was knocked-down in the identical setup as used here (Steiger et al 2008): Max-depletion in Myc-overexpressing eyes abrogated Myc-induced eye overgrowth, but did not affect Myc-induced apoptosis, thus resulting in smaller, more deformed eyes.

5. Figure 2A. (i) What are the black dots represent? (ii) n=6-109 - is it correct?

There was a mixup, for which we apologize. This entire experiment has been carried out twice. The old Fig 2A shows the result of one repeat, but the numbers 6-109 and the BLISS-score in the text refer to the second repeat. We have now replaced Fig 2A with a representation of this second repeat (for which the numbers 6 -109 are correct).

6. Figure S2A. The similar colors are very confusing. The authors should choose different colors.

The colors have been changed.

7. Figure 3C. It does not mention what exactly the luciferase activity reflects.

Fig 3C and its legend were changed to state "Relative number of NB II-derived cells", and the legend also points to the Methods, where the assay is explained. Briefly: luciferase expression is driven by the "wor-GAL4 ase-GAL80" system in NB II, hence luciferase activity reflects the number of NB II- derived cells (see Neumüller et al 2013 where this system was first described, Herter et al 2015 where we used this system in our labs).

8. Page 4, line 32: The authors should add information indicating what Deadpan (Dpn) and Asense (Ase) are.

We have added the following explanation to the text: "...stainings for Deadpan (Dpn) and Asense (Ase), which are transcription factors that serve as markers for neuroblast development: NB II (of which there are 8 per brain hemisphere) express Dpn but not Ase (Dpn+ Ase-), in contrast to type I NBs where both proteins are present (Dpn+ Ase+). NB II then generate intermediate neural progenitors (INPs) which pass through a maturation process, from Dpn- Ase- to Dpn- Ase+ to Dpn+ Ase+, before producing ganglion mother cells".

9. Figures 3E & 3F. I find some contradiction between these two figures concerning Ctr / Spt5-KD. In 3E it seems that Spt5 KD led to higher EdU incorporation, opposite to the results in Figure 3F and of what is stated in the text (p. 5, lines 2-3). Please clarify.

Fig 3E shows a low magnification of larval brain hemispheres. The compact EdU signals correspond to the proliferation centers of the optic lobes and dispersed signals in the central part correspond to dividing type I and type II neuroblasts, ganglion mother cells and INPs. Depending on the orientation of the brain, "signal density" appears sometimes different. We targeted only type II neuroblast lineages (labeled in green) with Spt5- and/or brat-knockdown and these clusters are overwhelmingly EdU-negative after Spt5-knockdown (Fig. 3F). A sentence emphasizing this point has been added to the text.

10. Figure 4B. "ctr" should be "tum" (similar to 4D). Otherwise, it is misleading.

To avoid confusion, we have replaced the term "tum" throughout Fig 4 with either "brat-KD" or "brat-KD Spt5-KD".

11. Figure 5C. See comment no. 3. Especially there is no difference between black and dark gray.

We have tried to use more distinct colors, while maintaining an identical color scheme between panels 5A & 5C.

Reviewer #2

A major goal in cancer therapy is to be able to identify drugs that eliminate tumors while exhibiting minor effects on healthy tissues. Genes of the Myc family are among the most commonly up-regulated genes in human cancers, acting as potent oncogenes. The study uses *Drosophila* to test the hypothesis that targeting a partner of Myc could prevent its oncogenic activity and stop tumor growth while leaving normal cells unaltered.

First the authors demonstrate that during *Drosophila* development, Myc and Spt5 (a binding partner of Myc in mammals) genetically interact. Then they convincingly show using a *Drosophila* brain tumor model that the knockdown of Spt5 is sufficient to significantly alter tumor growth. Although Spt5 is important and necessary for development, the authors nicely show that silencing Spt5 systemically from late larval stages can extend the lifespan of tumor bearing flies.

This work is interesting as it provides a proof of concept that Myc partners can be targeted for cancer therapy to efficiently slow down or stop tumor growth while minimizing the impact on the organism.

The experiments are convincing and clearly presented and the manuscript is clearly written.

I provide here a set of questions and comments that could be addressed in new experiments or in the text and may help to clarify some points and link the transcriptomic data to some of the phenotypes.

1. One question is whether Spt5-KD tumor cells are eliminated in the adult or remain silent (non-proliferative) as shown in larvae (Fig 3F). This could be tested by doing an anti-Dpn and anti-PH3 staining in bratKD / Spt5-KD adult flies.

+ 2. Fig 5A suggests that the rescue is not complete in brat-KD / Spt5-KD animals as they still die significantly earlier than Ctr / Spt5-KD. Is it because of a "relapse". Do they die because the tumor ultimately continues growing?

To address these 2 points, we have measured EdU incorporation in the brains of 3-, 10- and 20-days-old adult flies. Indeed, cells retained proliferation activity already at day 3 and formed clusters, albeit clearly smaller ones than in brat-KD brains. These proliferating cells ultimately formed large tumor masses at day 20, which is the likely reason for animal death. We have added these data to the text and as novel Fig 6. However, we do not know at present whether these Spt5-depleted cells have acquired the ability to proliferate with reduced levels of Spt5 (it is conceivable that the hormonal environment in adults imposes different constraints on cell proliferation than that in larvae), or whether Spt5 levels have rebounded in these "Spt5-depleted NB II" (e.g. because the levels of GAL4, driving the expression of siSpt5, decreased during the development of these cells)

3. It has been shown that the growth and maintenance of bratKD tumors rely on a specific sub-population of tumor cells, expressing Imp, that act as cancer stem cells (PMID: 27296804; PMID: 31566561). The transcriptomic analysis suggests that Imp is down-regulated in bratKD / Spt5-KD. It could be easily tested if this population is eliminated in this context using immunostaining, explaining why tumors stop growing.

+ 4. On the other hand, if Imp+ tumor cells remain in adults but in a dormant state, it could explain a possible relapse later.

+ 5. Does Spt5-OE expand the subpopulation of Imp+ cells in tumors, making them growing faster?

These are excellent questions. To address them, we have stained larval and adult brains for Imp (and Chinmo) and included these results in the text and as new Fig S5. Briefly: Spt5-KD enriched for Imp-/Chinmo-positive tumor cells. After adult eclosion these cells re-entered the cycle, and they also spawned Imp-/Chinmo-negative subpopulations, resulting in tumor heterogeneity akin to what has previously been reported by Genovese et al. 2019.

Unfortunately, the experiment addressing Spt5-OE technically failed. However, considering the comparatively small longevity difference between brat-KD tumors and Spt5-OE/brat-KD tumors, we might expect it difficult to detect quantitative differences in relative abundance of Imp-/Chinmo-positive cells between these genotypes.

6. What is the effect of silencing Myc in normal type-II lineages and in bratKD tumors? Are tumors eliminated?

As shown in Fig S4B, knockdown of Myc in animals carrying brat-KD-induced tumors extends their longevity, whereas the same Myc-knockdown in control NB II reduces viability of adults, with this effect becoming clear by 39 days after adult eclosion.

To assess the mass of NB II-derived tumors (or NB II-derived normal tissue) in response to Myc-knockdown, we rely on a luciferase transgene that is specifically expressed in NB II. Published measurements that were performed on L3 larvae (Neumüller et al 2013) or on <1-day-old adults (Herter et al 2015) show that Myc-KD only marginally reduces the mass/amount of clones derived from normal NB II (Herter et al 2015, Fig 5E: ca. -5%, $p > 0.05$), but that it significantly decreases (but does not eliminate) the tumor mass derived from brat-KD NB II (according to Herter et al. 2015, Fig 5E: -20%, $p < 0.005$; according to Neumüller et al 2013, Figs 5E & S8B: -50%, $p < 0.001$).

The following text (referring to these data) has been added: "... this is consistent with the published reduction of NB II tumor mass by Myc-knockdown (Neumüller et al 2013, Herter et al 2015)."

We can only speculate that the low level of NB II loss (the ~5% measured in 1-day-old adults) accumulates over time to interfere with survival at later time points, or that the quality of the NB II derived tissue remaining in 39-days old Myc-KD adults is altered such as to impair their survival.

7. The authors claim that they can systemically target Spt5 using the act-FRT-stop-FRT-siSpt5 transgene. It is unclear how efficient the hs-FLP-mediated recombination is between FRT sites in their assay. Is it possible to assess this using anti-Spt5 antibody? Can the lifespan be further extended if two or three heat-shocks are performed in order to increase the number of cells that have undergone the recombination event ?

This is a good suggestion. Since we could not obtain an anti-Spt5 antibody, we measured Spt5 transcript levels instead and found this direct system to be less efficient in depleting Spt5 than the GAL4-dependent indirect system – see Reviewer 1 Comment 1 above:

"Second, to independently evaluate our Spt5 transgenes, we induced ubiquitous expression of the Spt5-cDNA or of siSpt5 by a 1-hour heat-shock at 37 °C and analysed mRNA levels 1 day later in wandering larvae; as controls we used the corresponding genotypes without heat-shock. This resulted in the following expression levels:

for "hs-FLP actin5C-FRT-stop-FRT-GAL4 UAS-Spt5" (Spt5 overexpression): 750% (+/- 90%) of control;

for "hs-FLP actin5C-FRT-stop-FRT-GAL4 UAS-siSpt5" (the KD-transgene used for all experiments except Fig 5C): 62% (+/- 24%) of control;

for "hs-FLP actin5C-FRT-stop-FRT-siSpt5" (the KD-transgene used for Fig5C): 88% (+/- 24%) of control.

These values are now added to the text."

We don't know whether this is due to inefficient flip-out, or because the GAL4-system involves an amplification which results in higher levels of siRNA (our preferred explanation).

Unfortunately, time constraints for the revision precluded us from conducting a full survival experiment.

October 10, 2023

RE: Life Science Alliance Manuscript #LSA-2023-02130-TR

Dr. Peter Gallant
University of Würzburg
Department of Biochemistry and Molecular Biology, Theodor Boveri Institute, Biocenter
Am Hubland
Würzburg, Bavaria 97074
Germany

Dear Dr. Gallant,

Thank you for submitting your revised manuscript entitled "Spt5 interacts genetically with Myc and is limiting for brain tumor growth in *Drosophila*". We would be happy to publish your paper in Life Science Alliance pending final revisions necessary to meet our formatting guidelines.

- please add ORCID ID for secondary and third corresponding authors--they should have received instructions on how to do so
- please add Keywords for your manuscript to our system
- please add the Twitter handle of your host institute/organization as well as your own or/and one of the authors in our system
- please consult our manuscript preparation guidelines <https://www.life-science-alliance.org/manuscript-prep> and make sure your manuscript sections are in the correct order
- we encourage you to revise the figure legend for Figure 6 such that the figure panels are introduced in an alphabetical order
- please add your main, supplementary figure, and table legends to the main manuscript text after the references section
- please add a callout for Figure S2A to your main manuscript text

A. FINAL FILES:

B. MANUSCRIPT ORGANIZATION AND FORMATTING:

Sincerely,

Reviewer #1 (Comments to the Authors (Required)):

The revised manuscript addressed well all the concerns and is suitable for publication

October 13, 2023

RE: Life Science Alliance Manuscript #LSA-2023-02130-TRR

Dr. Peter Gallant
University of Würzburg
Department of Biochemistry and Molecular Biology, Theodor Boveri Institute, Biocenter
Am Hubland
Würzburg, Bavaria 97074
Germany

Dear Dr. Gallant,

Thank you for submitting your Research Article entitled "Spt5 interacts genetically with Myc and is limiting for brain tumor growth in *Drosophila*". It is a pleasure to let you know that your manuscript is now accepted for publication in Life Science Alliance. Congratulations on this interesting work.

DISTRIBUTION OF MATERIALS:

Again, congratulations on a very nice paper. I hope you found the review process to be constructive and are pleased with how the manuscript was handled editorially. We look forward to future exciting submissions from your lab.

Sincerely,
